# Magnesium-to-Calcium Ratio and Mortality from COVID-19

**DOI:** 10.3390/nu14091686

**Published:** 2022-04-19

**Authors:** Fernando Guerrero-Romero, Moises Mercado, Martha Rodríguez-Morán, Claudia Ramírez-Renteria, Gerardo Martínez-Aguilar, Daniel Marrero-Rodríguez, Aldo Ferreira-Hermosillo, Luis E. Simental-Mendía, Ilan Remba-Shapiro, Claudia I. Gamboa-Gómez, Alejandra Albarrán-Sánchez, Miriam L. Sanchez-García

**Affiliations:** 1Biomedical Research Unit, Instituto Mexicano del Seguro Social, Durango 34067, Mexico; rodriguez.moran.martha@gmail.com (M.R.-M.); uimec@yahoo.es (G.M.-A.); luis_simental81@hotmail.com (L.E.S.-M.); clau140382@hotmail.com (C.I.G.-G.); 2Research Unit in Endocrine Diseases, Hospital de Especialidades, Centro Médico Nacional Siglo XXI, Instituto Mexicano del Seguro Social, Mexico City 06720, Mexico; mmercadoa@yahoo.com (M.M.); clau.r2000@gmail.com (C.R.-R.); dan.mar57@gmail.com (D.M.-R.); aldo.nagisa@gmail.com (A.F.-H.); ilanremba@gmail.com (I.R.-S.); mrmsaga@gmail.com (M.L.S.-G.); 3Department of Internal Medicine, Hospital de Especialidades, Centro Médico Nacional Siglo XXI, Instituto Mexicano del Seguro Social, Mexico City 06720, Mexico; albarranalejandra@gmail.com

**Keywords:** COVID-19, magnesium-to-calcium ratio, mortality

## Abstract

Obesity, type 2 diabetes, arterial hypertension, decrease in immune response, cytokine storm, endothelial dysfunction, and arrhythmias, which are frequent in COVID-19 patients, are associated with hypomagnesemia. Given that cellular influx and efflux of magnesium and calcium involve the same transporters, we aimed to evaluate the association of serum magnesium-to-calcium ratio with mortality from severe COVID-19. The clinical and laboratory data of 1064 patients, aged 60.3 ± 15.7 years, and hospitalized by COVID-19 from March 2020 to July 2021 were analyzed. The data of 554 (52%) patients discharged per death were compared with the data of 510 (48%) patients discharged per recovery. The ROC curve showed that the best cut-off point of the magnesium-to-calcium ratio for identifying individuals at high risk of mortality from COVID-19 was 0.20. The sensitivity and specificity were 83% and 24%. The adjusted multivariate regression model showed that the odds ratio between the magnesium-to-calcium ratio ≤0.20 and discharge per death from COVID-19 was 6.93 (_95%_CI 1.6–29.1) in the whole population, 4.93 (_95%_CI 1.4–19.1, *p* = 0.003) in men, and 3.93 (_95%_CI 1.6–9.3) in women. In conclusion, our results show that a magnesium-to-calcium ratio ≤0.20 is strongly associated with mortality in patients with severe COVID-19.

## 1. Introduction

Coronavirus disease-19 (COVID-19), an acute respiratory syndrome due to infection with SARS-CoV-2, has been rapidly spreading worldwide [1,2]. The clinical spectrum of COVID-19 ranges from an asymptomatic carrier state, or a mild upper respiratory tract infection, to a devastating acute respiratory distress syndrome with multiple organ failure and high mortality rate [3].

Although several risk factors for developing the severe form of the disease have been identified, the development of biomarkers to predict poor outcome and/or mortality from COVID-19 remains a requirement.

Interestingly, several of the well-known COVID-19 risk factors and some co-morbidities such as acute renal failure, arterial and venous thrombosis, cardiac failure, and arrhythmias are linked to magnesium deficiency [4,5]. Therefore, we hypothesized that hypomagnesemia might play an important role in the pathophysiology and mortality from COVID-19 [5,6].

Hypomagnesemia is associated with a decrease synthesis and activation in vitamin D, an increase in oxidative stress and cytotoxic activity of T lymphocytes, and with the triggering of cytokine storm [7,8]. Hypomagnesemia is also related to abnormal platelet aggregation, coagulation abnormalities [9,10], endothelial dysfunction [11], and myocardial damage [12], entities frequently identified in severely ill COVID-19 patients [5,13,14].

Calcium is involved in several pathophysiological aspects of the interaction between the SARS-CoV-2 and human host cell [15,16,17]. With regard to this, the association between hypocalcemia and the clinical severity and mortality by COVID-19 has been consistently reported [18,19]. Thus, low serum calcium levels are considered as prognostic factors in determining the severity of the disease [20].

Given that magnesium is a calcium antagonist and that cellular influx and efflux of magnesium and calcium involve the same transporters [21,22], the functions of both cations are closely linked. Thus, the aim of this study was to evaluate the association of serum magnesium-to-calcium ratio with mortality from severe COVID-19.

## 2. Materials and Methods

A cross-sectional retrospective analysis was carried out. Clinical and laboratory data were collected from the medical records of 1616 patients hospitalized from March 2020 to July 2021 in a tertiary care hospital, in Mexico City, and a general hospital, in Durango City, in central and northern Mexico, respectively.

The study was approved by the Institutional Review Board and Ethical Committee of the Mexican Social Security Institute (R-2021-785-076, 4 August 2021).

Eligible participants were individuals older than 20 years with a positive polymerase chain reaction (PCR) test for SARS-CoV-2 and a diagnosis of severe COVID-19.

Missing laboratory data, previous chronic renal disease or creatinine levels >1.2 mg/dL, heart or liver failure, cancer, and stroke were exclusion criteria. Patients with a history of having received calcium or magnesium supplements, or glucocorticoids over the last month, or intravenous fluids before their initial evaluation, were also excluded.

Data from patients discharged per death were compared vs. data from patients discharged per recovery.

According to the international epidemiologic criteria for COVID-19 in adults, severe COVID-19 was defined by the presence of dyspnea, respiratory rate ≥30 breaths per minute, oxygen saturation ≤93%, partial pressure of arterial oxygen/fraction of inspired oxygen ratio <300 mmHg, and/or lung infiltrates involving ≥50% [23].

The magnesium-to-calcium ratio was calculated as total serum magnesium levels (mg/dL)/calcium (mg/dL).

Upon admission, blood samples were collected from the antecubital vein before any therapy or intravenous fluids were started. Laboratory data included cell blood count with differentials, prothrombin and partial thromboplastin times, serum magnesium, calcium, albumin, fasting glucose, creatinine, liver function tests, D-dimer, and C-Reactive protein levels.

### Statistical Analysis

Normally distributed numerical values are reported as mean ± standard deviation and non-normally distributed values as median with interquartile range. Data distribution was ascertained by mean of the Shapiro–Wilk test.

Bivariate analysis was performed using unpaired Student’s t-test (Mann–Whitney U test for skewed data) or chi-square test.

The best cut-off point of the magnesium-to-calcium ratio, as a biomarker associated with mortality from COVID-19, as well as its specificity and sensitivity, were estimated in a receiver-operating characteristic (ROC) curve.

A multiple logistic regression analysis was conducted in order to determine the odds ratio (OR) between the magnesium-to-calcium ratio (independent variable) and mortality from COVID-19 (dependent variable). In order to control the potential confounders, age, hemoglobin, leukocyte, neutrophils, lymphocytes, fasting glucose, albumin, and hsCRP were introduced, as continuous variables, in an additional logistic regression analysis. In the stratified analysis by sex, in addition to the above-mentioned variables, the logistic regression model was also adjusted by diabetes, hypertension, obesity, and creatinine. All these were the variables that in the bivariate analysis showed significant statistical differences; see Table 1 and Table 2.

Data were analyzed using the statistical package SPSS version 21.0 (SPSS Inc., Chicago, IL, USA). A 95% confidence interval (_95%_CI) or a *p* value < 0.05 defined the statistical significance.

## 3. Results

A total of 1616 individuals were screened; of these, 552 (34.1%) were excluded by the presence of exclusion criteria. Hence, data from 1064 patients, with mean age of 60.3 ± 15.7 years, were analyzed; see Figure 1.

The proportion of obesity, diabetes, hypertension, and chronic obstructive pulmonary disease (COPD) was 36.9%, 40.5%, 45.4%, 4.9%, respectively, without significant differences between the groups; see Table 1.

Individuals discharged per death, as compared with those discharged per recovery, were older and showed higher total leukocytes and neutrophils counts, as well as higher glucose, D-dimer, and hsCRP levels. Furthermore, hemoglobin levels, lymphocytes counts, serum albumin and magnesium levels, and the magnesium-to-calcium ratio were lower in the patients discharged per death; see Table 1

Stratified analysis by sex is shown in Table 2. Men and women discharged per death were older, exhibited higher leukocytes and neutrophils counts, higher serum calcium and hsCRP levels, and lower lymphocytes counts, serum albumin and magnesium than patients discharged per recovery.

The proportion of hypomagnesemia (54.3% vs. 32.9%, *p* = 0.002) and hypocalcemia (16.4% vs. 9.5%, *p* = 0.01) was significantly higher in the patients discharged per death than in those discharged per recovery.

A total of 262 (24.6%) individuals required admission to the Intensive Care Unit; of those, 156 (28.1%) and 106 (20.8%) were in the groups of patients discharged per death and per recovery (*p* = 0.006).

The proportion of individuals who exhibited a magnesium to-calcium ratio ≤0.20 was significantly higher in the individuals discharged per death than in those discharged by recovery (93.5% vs. 14.7%, respectively, *p* = 0.0001).

In the whole population (R −0.135, *p* = 0.001), as well as in women (R −0.124, *p* = 0.04) and men (R −0.142, *p* = 0.009), the magnesium-to-calcium ratio was inversely correlated with the discharge per death.

The ROC curve showed that the best cut-off point of the magnesium-to-calcium ratio for identifying individuals at high risk of mortality from COVID-19 was 0.20. The value of the sensitivity, specificity and area under the curve was 83%, 24%, and 0.521, respectively; see Figure 2.

The non-adjusted multivariate logistic regression analysis showed a significant association between the magnesium-to-calcium ratio ≤0.20 and discharge per death, in the whole population (OR 4.0; _95%_CI 2.2–7.3, *p* = 0.0001), in men (OR 4.5; _95%_CI 1.6–12.0, *p* = 0.001), and women (OR 3.6; _95%_CI 1.7–7.9, *p* = 0.001). In the adjusted regression model, the magnesium-to-calcium ratio ≤ 0.20 remained significantly associated with discharge per death in the whole population (OR 6.93, _95%_CI 1.6–29.1), in men (OR 4.93, _95%_CI 1.4–19.1, *p* = 0.003), and women (OR 3.93, _95%_CI 1.6–9.3).

## 4. Discussion

Our study showing that the magnesium-to-calcium ratio ≤ 0.20 is strongly associated with mortality from severe COVID-19 provides a new insight that could be useful in the management of the disease.

This ratio has been used as a mortality predictor in patients with cardiovascular and neoplastic diseases [24,25,26]. However, to the best of our knowledge there are no previous reports evaluating the calcium-to-magnesium ratio as a biomarker for mortality from COVID-19.

Given the elevated rates of mortality from COVID-19, there is an urgent need to identify those patients at high risk of death. Monitoring serum calcium and magnesium levels during hospitalization may represent a useful strategy for the early identification of high-risk patients, which would allow timely treatment and appropriate surveillance. Although it remains to be proven whether adding supplementation with magnesium and/or calcium to a treatment for COVID-19 can reduce the likelihood of death, restoring a magnesium-calcium balance is mandatory.

Given that measuring serum calcium and magnesium is low-cost and is available in every hospital laboratory, monitoring these cations could be a cost-effective strategy for identifying the high-risk patients.

In our study, patients discharged per death, as compared with those discharged per recovery, were older without significant differences regarding other co-morbidities related with poor outcome from COVID-19. Patients who died exhibited higher fasting glucose, hsCRP levels, leukocytes and neutrophils, as well as lower albumin, hemoglobin and lymphocytes; parameters that have important prognostic implications. With this in mind, it has been reported that older age, anemia, hypoalbuminemia, lymphopenia, and hyperglycemia, regardless of inflammatory markers, are associated with in-hospital mortality from COVID-19 [27,28,29,30,31]. In addition, it is well known that biomarkers of inflammation and the high neutrophil-to-lymphocyte ratio are independently associated with severe COVID-19 and mortality from COVID-19 [32,33]. In order to control the potential source of bias related with the risk factors mentioned above, the logistic regression model was adjusted for all these variables. In the adjusted model, the calcium-to-magnesium ratio ≤ 0.20 remained significantly associated with mortality from COVID-19. This finding emphasizes the important role that the magnesium-calcium balance may play in the prognosis of COVID-19.

Furthermore, patients discharged per death had lower serum magnesium levels, a finding suggesting that hypomagnesemia could be a biomarker of a poor outcome from COVID-19. Findings supported the hypothesis by Iotti et al. [34], which states that a low magnesium status might be related to the critical clinical manifestations of COVID-19. With regard to this, hypomagnesemia has been found to be inversely associated with cardiovascular disease and chronic inflammation [35], entities related to a poor outcome from COVID-19.

The magnesium-to-calcium ratio exhibited high sensitivity (83%) with low specificity (24%). Given that sensitivity is the proportion of individuals who have an event and a positive test result, tests with high sensitivity are useful for ruling out a disease if a person tests negative [36]. Thus, tests with high sensitivity are appropriate for the screening of individuals at high risk. With regard to this, the magnesium-to-calcium ratio exhibiting high sensitivity would be suitable for detecting individuals at high risk of mortality from COVID-19.

Finally, we did not measure arterial blood gases for all patients; thus, we could not adjust the logistic regression analysis by the gasometry parameters. Undoubtedly, this is a limitation of the study. The lack of pro-inflammatory cytokines measurements that did not allow us to entirely assess the inflammatory response to COVID-19 and the cross-sectional design used are also among the limitations of our study. Further research for a better understanding of the pathophysiological pathways involved in the adverse evolution of COVID-19 is required.

In conclusion, our results show that the magnesium-to-calcium ratio ≤ 0.20 is strongly associated with mortality from severe COVID-19.

## Figures and Tables

**Figure 1 nutrients-14-01686-f001:**
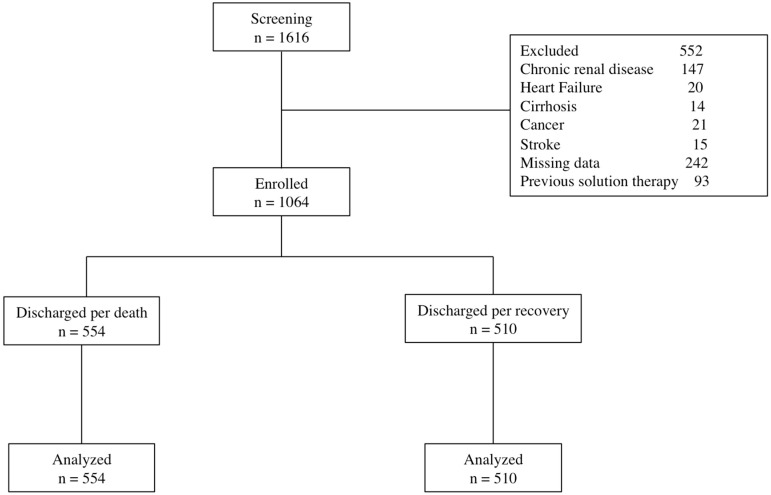
Flow chart.

**Figure 2 nutrients-14-01686-f002:**
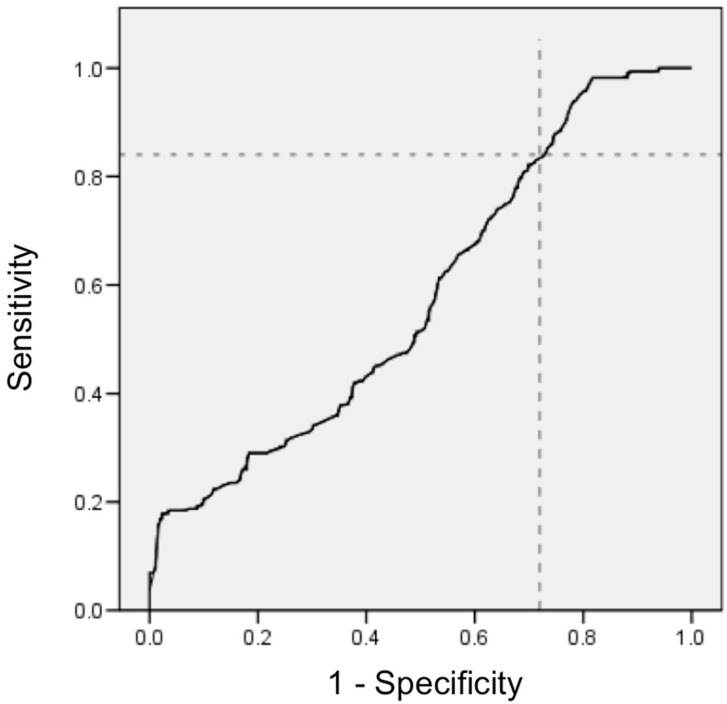
ROC curve plotting the sensitivity and specificity of the magnesium-to-calcium ratio and mortality from severe COVID-19.

**Table 1 nutrients-14-01686-t001:** Clinical and biochemical characteristics of the target population, N = 1064.

	Hospital Discharged		
	Death	Recovery	
N	554	510	*p* Value
Age, years	64.4 ± 14.7	57.5 ± 15.8	0.0001
Oxygen saturation, %	85.2 ± 5.9	89.4 ± 1.2	0.16
Body mass index, kg/m^2^	29.9 ± 7.3	29.6 ± 7.1	0.66
Obesity, n (%)	209 (37.7)	184 (36.1)	0.62
Diabetes, n (%)	221 (39.9)	201 (39.4)	0.92
Hypertension, n (%)	260 (46.9)	223 (43.7)	0.32
Chronic obstructive pulmonary disease, n (%)	19 (3.4)	31 (6.1)	0.22
Hemoglobin	13.6 ± 2.8	14.1 ± 2.4	0.02
Leukocyte	11.15 (8.0–16.0)	9.21 (0.68–12.6)	0.001
Neutrophils	9.8 (6.6–14.1)	7.35 (4.90–10.7)	0.0001
Lymphocytes	0.79 (0.54–1.21)	0.90 (0.74–1.45)	0.0001
Platelets	250 (185–336)	254 (195–326)	0.371
Fasting glucose, mg/dL	203.7 ± 156.1	163.1 ± 99.9	0.0001
Serum creatinine, mg/dL	1.7 ± 0.4	1.4 ± 0.5	0.08
Albumin, g/L	3.2 ± 0.6	3.5 ± 0.6	0.0001
Magnesium, mg/dL	1.91 ± 0.31	1.97 ± 0.23	0.01
Calcium, mg/dL	8.3 ± 0.9	8.1 ± 1.2	0.39
Magnesium/calcium ratio	0.23 ± 0.05	0.26 ± 0.10	0.02
D-dimer, mg/dL ^†^	0.94 (0.31–0.84)	0.66 (0.32–0.81)	0.37
hsC-reactive protein	21.6 (8.2–31.4)	12.9 (5.5–24.1)	0.0001

^†^ Values are median (25th–75th).

**Table 2 nutrients-14-01686-t002:** Characteristics of the target population, N = 1064.

	Women			Men		
	Hospital Discharged			Hospital Discharged		
	Death	Recovery		Death	Recovery	
N	276	225	*p* Value	278	285	*p* Value
Age, years	65.6 ± 13.8	59.01 ± 15.5	0.001	63.5 ± 15.0	55.3 ± 15.9	0.0001
Diabetes, n (%)	91 (32.9)	139 (61.8)	0.0001	96 (34.5)	95 (33.3)	0.83
Hypertension	112 (40.6)	135 (60.6)	0.0002	118 (42.4)	116 (40.7)	0.73
Obesity, n (%)	84 (30.4)	124 (55.1)	0.0001	71 (25.5)	114 (40.0)	0.0003
CPOD *, n (%)	9 (3.3)	18 (8.0)	0.41	10 (3.6)	12 (4.2)	0.95
Body mass index, kg/m^2^	31.2 ± 7.5	30.8 ± 7.7	0.63	28.8 ± 7.1	28.6 ± 6.5	0.67
Hemoglobin,	12.7 ± 2.6	13.2 ± 2.2	0.09	14.1 ± 2.7	14.8 ± 2.3	0.03
Leukocytes ^†^	10.9 (7.7–14.7)	9.10 (6.6–12.1)	0.001	11.26 (8.2–16.2)	9.60 (7.1–13.9)	0.002
Neutrophils ^†^	9.3 (6.4–12.3)	6.98 (4.7–9.66)	0.001	10.22 (6.7–14.2)	7.6 (5.1–11.6)	0.0001
Lymphocytes ^†^	0.95 (0.57–1.42)	1.11 (0.81–1.67)	0.03	0.74 (0.52–0.99)	0.98 (0.69–1.37)	0.001
Platelets	252 (182–339)	281 (204–340)	0.43	247 (192–322)	239 (187–317)	0.67
Fasting glucose, mg/dL	209.7 ± 161.2	174.9 ± 110.6	0.06	200.8 ± 154.0	153.8 ± 92.0	0.001
Creatinine, mg/dL	2.1 ± 0.3	1.4 ± 0.2	0.03	1.99 ± 0.3	1.65 ± 0. 3	0.24
Albumin, g/L	3.2 ± 0.5	3.6 ± 0.6	0.001	3.3 ± 0.7	3.5 ± 0.6	0.001
Serum Calcium, mg/dL	8.1± 1.2	7.5 ± 2.1	0.002	8.4 ± 0.7	8.1 ± 0.9	0.01
Serum Magnesium, mg/dL	1.90 ± 0.29	1.98 ± 0.24	0.04	1.91 ± 0.32	2.01 ± 0.22	0.006
Magnesium/calcium ratio	0.24 ± 0.03	0.28 ± 0.01	0.0001	0.23 ± 0.04	0.26 ± 0.01	0.0001
D-dimer, mg/dL ^†^	1.34 (0.53–5.24)	0.43 (0.33–0.81)	0.38	0.60 (0.28–2.10)	0.48 (0.28–0.87)	0.43
C-reactive protein ^†^	20.1 (6.8–27.4)	9.0 (5.5–21.9)	0.008	21.9 (9.0–20.7)	16.0 (5.5–26.1)	0.0001

* Chronic Obstructive Pulmonary Disease; Data are mean ± SD, otherwise is indicated; ^†^ Values are median (25th–75th).

## Data Availability

Data sharing would be available via prior request and with the knowledge and approval of the Institutional Ethical Committee.

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
