# Peer review of "Magnesium-to-Calcium Ratio and Mortality from COVID-19"

_nutrients, 2022, doi:10.3390/nu14091686_

Round 1

Reviewer 1 Report

This study investigated the association of serum magnesium-to-calcium ratio with mortality by severe COVID-19. For the study, data of 1,064 patients hospitalized for COVID-19 were collected and analyzed. The data of the patients discharged per death were compared with the data of the patients discharged per recovery. The ROC curve showed that the best cutoff point of the Magnesium-to-Calcium ratio for identifying individuals at high risk of mortality by COVID-19 was 0.20. The sensitivity and Specificity was 83% and 24%. The adjusted multivariate regression model showed that the odds ratio between the magnesium-to-calcium ratio ≤0.20 and discharge per death by COVID-19 was 6.93. The conclusion of the authors, supported by the data presented in the manuscript, is that the magnesium-to-calcium ratio ≤0.20 is strongly associated with mortality in patients with severe COVID-19.

Reviewer 2 Report

The paper  is well written and very clear. An undoubted value of this study is the large cohort of patients and the great variety of parameters measured. For this reason the discussion must be integrated, as  in some points it appears too concise essential.

Minor revisions:

1.The authors write: Men and women discharged per death were older, exhibited higher leukocytes and neutrophils count, higher serum calcium and hsCRP levels, and lower lymphocytes count, serum albumin and magnesium than patients discharged per recovery (lines 116-119)

There is a rich literature on the relationship between these blood parameters and hypomagnesaemia. Moreover, several studies have already associated changes of these parameters to Covid-19. (e.g., Iotti S. et al Magnes Res 2020; 33(2): 21)  A more in-depth discussion is desirable and recommended, taking into account the most recent work in this area.

2.The authors assert: we could not appropriately evaluate the association between calcium-to-magnesium ratio and gasometry parameters (lines 193-194)

Why put a table of values that are not related to the focus of the study? Furthermore, the reported values do not even vary among the cohort of patients discharged for death or recovery. Please include short discussion in relation to serum magnesium and calcium or remove that table.

Reviewer 3 Report

The authors investigate that magnesium-to-calcium ratio and mortality by COVID-19. The major findings are that magnisum-to-calcium ratio £ 0.20 is strongly associated with mortality in patients with severe COVID-19. This article has some concerns to be addressed as follows.

The major concerns:

  1. In methodology, a multiple regression analysis was conducted in order to determine the OR between the magnism-to-calcium ratio and mortality by COVID-19. How many and what kinds of potential confounders were analyzed? Can authors describe the detail elements and data in clinical and biochemical characteristics and process more clearly, in page 8, line 145-151?
  2. Given that measuring serum calcium and magnesium has low-cost and is available in every hospital laboratories, did author monitoring this parameter in a sequential way to see if there is another potential predictor for mortality in severe patients with COVID-19 infection.

Round 2

Reviewer 3 Report

I just curious about the issue that in page 9, the sentence of "Funding: Consejo Estatal de Ciencia y Tecnología del Estado de Durango (Folio 488)." was added. Why is it added in revision article? I can not make out whether there exist any probability of "Conflict of Interest"? Please explain it with the goal of "no conflict of interest" between this article and the funding. 
